# Efficacy of mecobalamin (vitamin B$_{12}$) in the treatment of long-term pain in women diagnosed with fibromyalgia: protocol for a randomised, placebo-controlled trial

Karin Sall Hansson,[1,2] Gunilla Lindqvist,[1] Kent Stening,[1] Jan Fohlman,[3] Anna Wojanowski,[3] Moa Ponten,[4] Karin Jensen,[4] Björn Gerdle,[5] Carina Elmqvist [1,2,3]

For numbered affiliations see end of article.

**Correspondence to**
Dr Carina Elmqvist;
carina.elmqvist@lnu.se

## ABSTRACT

**Introduction** Fibromyalgia causes long-term pain. It affects at least 2% of the population, the majority being women. In addition, extended symptoms corresponding to vitamin B$_{12}$ deficiency occur. Findings from several studies have indicated that vitamin B$_{12}$ may be a possible treatment for pain in fibromyalgia. The aim of the proposed study is to evaluate whether vitamin B$_{12}$ decreases pain sensitivity and the experience of pain (ie, hyperalgesia and allodynia) in women with fibromyalgia.

**Methods and analysis** The study is a randomised, placebo-controlled, single-blind, clinical trial with two parallel groups which are administered mecobalamin (vitamin B$_{12}$) or placebo over 12 weeks. 40 Swedish women aged 20–70 years with an earlier recorded diagnosis of fibromyalgia are randomised into the placebo group or the treatment group, each consisting of 20 participants. Outcomes consist of questionnaires measured at baseline and after 12 weeks of treatment. A final re-evaluation will then follow 12 weeks after treatment ends. The primary outcome is tolerance time, maximised to 3 min, which is assessed using the cold pressor test. In order to broaden the understanding of the lived experience of participants, qualitative interviews will be conducted using a phenomenological approach on a lifeworld theoretical basis (reflective lifeworld research approach).

**Ethics and dissemination** The protocol for the study is approved by the local ethical committee at Linkoping (EPM; 2018/294–31, appendices 2019–00347 and 2020–04482). The principles of the Helsinki Declaration are followed regarding oral and written consent to participate, confidentiality and the possibility to withdraw participation from the study at any time. The results will primarily be communicated through peer-reviewed journals and conferences.

**Trial registration number** NCT05008042.

## STRENGTHS AND LIMITATIONS OF THIS STUDY

⇒ The study design is a randomised, placebo-controlled, single-blind clinical trial using parallel groups.
⇒ The trial is a collaboration between different universities and regions.
⇒ A possible limitation is the low number of participants; however, the study continues until 20 participants per group have been included and completed the study, which exceeds the minimum number of 16 determined in our power calculation.
⇒ Another possible limitation is heterogeneity of the fibromyalgia spectrum.
⇒ Another possible limitation is the placebo effects of parenteral injection; however, the participants are individually randomised to the intervention or the control group (placebo) in a 1:1 allocation ratio.

## INTRODUCTION

Fibromyalgia causes long-term pain and affects at least 2% of the population, with 80% of the sufferers being women.[1] Treatment options are scarce and patients with fibromyalgia have experienced being discredited and invalidated by the healthcare system.[2] Fibromyalgia is characterised by long-term, widespread musculoskeletal pain and generalised hyperalgesia. This is often accompanied by fatigue, concentration problems and sleep problems.[3]

Fibromyalgia pain is currently classified as nociplastic, which means that pain arises from altered regulation of pain signals, in absence of tissue damage.[4] No definite pathophysiology has been established for fibromyalgia. Imaging techniques have challenged previous ideas about the peripheral origin of FM and have provided evidence for altered central nervous system nociceptive/pain processing and morphology in fibromyalgia. Furthermore, recent studies report both central alterations and peripheral alterations

(eg, systemic low-grade inflammation and nociceptor/muscle alterations).[5–9]

Current treatment for fibromyalgia includes both non-pharmacological and pharmacological interventions, depending on the key symptoms and the extent of disability. The recommended pharmacological treatments for fibromyalgia are antidepressants (eg, Serotonin and norepinephrine reuptake inhibitors (SNRI)) and antiepileptics (eg, gabapentinoids), which often cause side effects.[4] In addition, opioid analgetics and NSAID (Non-Steroidal Anti-Inflammatory Drugs)/COX (Cyclooxygenase inhibitor) are often prescribed.[10] Only a minority of individuals report a clinically relevant improvement from any of the treatments.[11]

Vitamin $B_{12}$ is sometimes used for symptoms other than those of vitamin $B_{12}$ deficiency, for example, different pain conditions such as backpain and neuropathic pain.[12–16] Vitamin $B_{12}$ nasal drops have been tested with positive results on patients with myalgic encephalomyelitis (ME)/chronic fatigue (CF)/fibromyalgia (ME/CF syndrome).[17] A recent study showed that sublingual vitamin $B_{12}$ had positive effect on patients with fibromyalgia.[18]

In addition, there are vitamin $B_{12}$ studies that have shown good results in the therapy of aphthous ulcers[19] and acute lumbago,[20] as well as in studies concerning diabetic polyneuropathy.[21 22] Moreover, studies examining methylcobalamin treatment in patients with lower back pain showed pain reduction and functionality gain.[13 23]

To summarise, it is not entirely clear how vitamin $B_{12}$ affects the human pain system. However, studies indicate that it may be a possible treatment in fibromyalgia.

### Aims and objectives

The aims of this study are to evaluate the effect of mecobalamin (vitamin $B_{12}$) and to describe lived experiences of pain, health, suffering and well-being in women with diagnosed fibromyalgia.

The primary objective is to evaluate whether mecobalamin (vitamin $B_{12}$), given as an intramuscular injection once a week for 12 weeks compared with placebo reduces pain sensitivity, that is, tolerance time (cold pressor test).

The secondary objective is to evaluate whether intramuscular mecobalamin (vitamin $B_{12}$) compared with placebo reduces pain intensity and pressure pain threshold (Numerical Rating Scale (NRS) and pressure algometry), and increases activity level (questionnaire), quality of life (questionnaire) and perceived effect of a given drug (questionnaire). Furthermore, the ratings of expected effects of a given drug (NRS), the desire for pain relief (NRS) and estimated pain variability (NRS) are evaluated. Qualitative interviews that describe how women with fibromyalgia experience pain, health, suffering and well-being (interviews) will be conducted.

## METHODS AND ANALYSIS

### Study design

The study is an academic randomised, placebo-controlled, single-blind clinical trial using parallel groups. Participants are individually randomised to intervention or control group (placebo) in a 1:1 allocation. In order to broaden and deepen the understanding of the lived experience, an interview will be conducted applying a phenomenological approach on a lifeworld theoretical basis, reflective lifeworld research (RLR) approach.[24]

### Study setting and sample

Swedish women aged 20–70 years with an earlier recorded diagnosed fibromyalgia. Detailed eligibility criteria are presented in table 1.

The participants are recruited via advertising in the local newspaper, Facebook, YouTube, Fibromyalgia Association, posters at Linnaeus University, hospitals and healthcare centres. The prospective participant contacts the trial manager via telephone or email and receives oral information. Written information and consent forms (online supplemental file 1) are sent to the participant, who contacts the trial manager for an appointment.

### Intervention and placebo

The active substance of vitamin $B_{12}$ given in the study is 2 mL mecobalamin (5 mg/ml), administered intramuscularly.[25] Placebo substance is 2 mL sodium chloride (9 mg/ml), isotonic solution for parenteral use (Baxter) intramuscularly.[26] All participants are informed that their urine may turn red, which may occur during mecobalamin injections.

### Measurements

At the first measurement opportunity, the trial manager and coexaminer 1 carefully follow inclusion and exclusion criteria. Oral information is given again, and consent is signed by the participant and by coexaminer 1. First,

| Table 1 | Inclusion and exclusion criteria |
|---|---|
| **Inclusion criteria** | **Exclusion criteria** |
| ► Diagnosis of fibromyalgia.<br>► Women aged 20–70 years.<br>► Swedish-speaking.<br>► Safe method of contraception.<br>► Cobalamin value >250 pmol/L < 800 pmol/L.<br>► Kidney function value (relatively) GFR >60 .mL/min/1.73 m².<br>► Liver function value.<br>– P-ALP 0.6–2.85 µkat/L.<br>– P-ALAT 0.15–1.13 µkat/L.<br>► Heart function value.<br>► Troponin-T <15 ng/L.<br>– NT-pro-BNP.<br>– <60 years <125 ng/L.<br>– NT-pro-BNP.<br>– 60 years <300.<br>– Given consent to participate. | ► Previous treatment with $B_{12}$.<br>► Known hypersensitivity to the active substance. mecobalamin or an additive.<br>► Neuroleptic treatment.<br>► Reynaud's phenomenon.<br>► Known neuropathy.<br>► Vegan as veganism can lead to $B_{12}$ deficiency.<br>► Known heart, kidney or liver disease.<br>► Breast feeding.<br>► Planned or ongoing pregnancy. |

the participant answers two questionnaires (short-form McGill Pain Questionnaire (MPQ)[27] and RAND-36 (RAND-36 Health-Related Quality of Life)),[28] then stimulation tests (cold pressor test and pressure algometry)[29–33] are performed followed by pain rating (NRS)[34] immediately after 1 min and after 3 min. Finally, blood samples for cobalamin (vitamin $B_{12}$), kidney, liver and heart markers are taken. The participant provides an information card about the study's design, purpose, treatment options and contacts with telephone numbers. The participant is asked about her interest in an interview in connection with the final measurement opportunity. Coexaminer 2 assesses the blood samples and approves the final inclusion. If tests deviate from accepted reference values, the participant is excluded and encouraged to seek medical attention.

After the approved inclusion, an independent chief examiner performs the randomisation. The participant then receives intramuscular injections of the active substance or placebo, once a week for 12 weeks, given by a registered nurse. Before each injection, the participant answers questions in which he or she estimate his or her expected effects of a given drug, desire for pain relief and average pain intensity during the past week (NRS). The participant has the opportunity to postpone the visit for ±3 days, for example, in case of illness or travelling and may miss a maximum of two injections in order to follow per-protocol analysis. Even if the participant is not compliant with the study protocol, the participant may retain the treatment regime for the duration of the study and may be analysed in the main intention-to-treat analysis.

The second measurement takes place after 12 weeks of intervention and three questionnaires (MPQ, RAND-36 and Patient Global Impression of Change (PGIC))[35] are filled out. The stimulation tests cold pressor test and pressure algometry are performed as well as blood sampling of cobalamin/p (vitamin $B_{12}$). In cases of low cobalamin/p, the participant is encouraged to seek medical attention but is still included in the study.

At the third measurement, which takes place 12 weeks after the end of the intervention, the participant fills in questionnaires (MPQ, RAND-36 and PGIC), undergoes stimulation tests (cold pressor test and pressure algometry), blood sampling of cobalamin/p (vitamin $B_{12}$) together with an interview. If cobalamin/p on this occasion shows that the participant has a low cobalamin value, the participant is encouraged to seek medical attention. For the individual participant, the study ends after 6 months. An overview of the study flow is presented in figure 1.

## Outcomes

The primary outcome is tolerance time, maximised to 3 min, which is tested using the cold pressor test.

The secondary outcomes are
▶ Pain experience measured by a pressure algometry test performed on the shoulder, hip, knee and elbow.
▶ Possible pain change measured by a pressure algometry test performed on the shoulder, hip, knee and elbow.
▶ Subjective experience of pain measured using NRS of 0–10, where 0 is the best outcome.

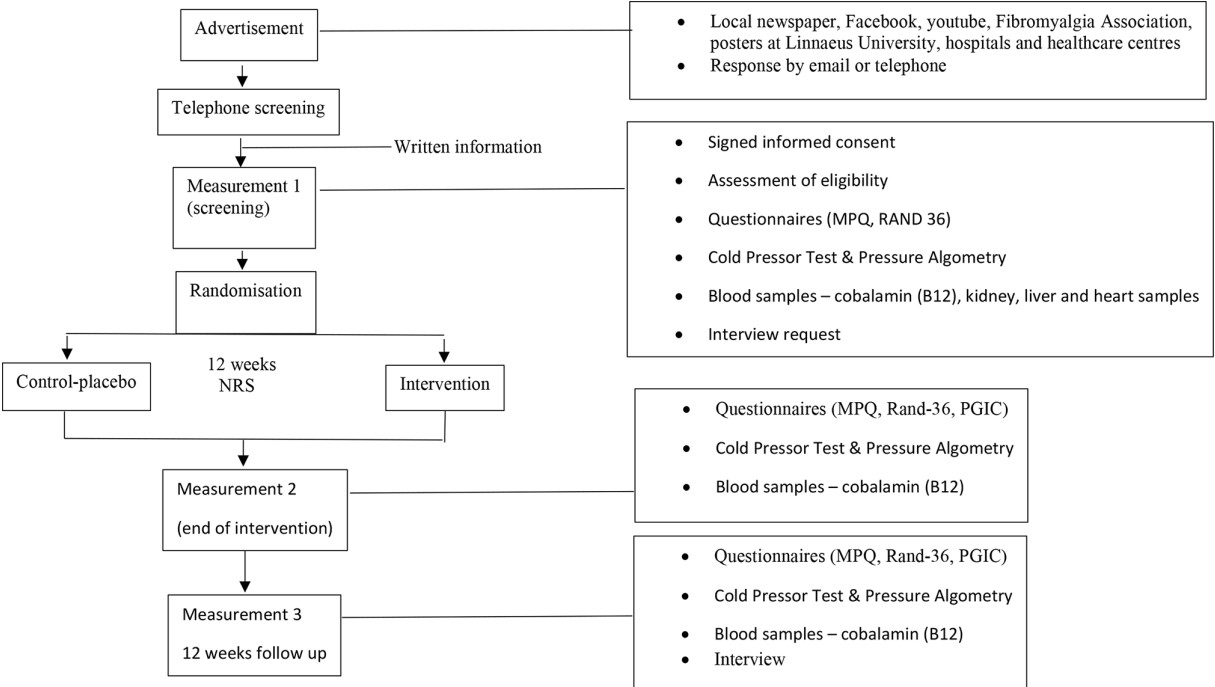

**Figure 1** Flowchart of participants, outcome measures and follow-up points. MPQ, McGill Pain Questionnaire; PGIC, Patient Global Impression of Change.

- Possible pain change measured using NRS of 0–10, where 0 is the best outcome.
- Ratings of expectation, desire for relief, using NRS of 0–10, where 0 is the best outcome.
- Ratings of expectation, pain variability, using NRS of 0–10, where 0 is the best outcome.
- Activity levels are assessed using MPQ Short Version score of 0–45.
- Quality of life is assessed using questionnaire RAND-36 score of 0–100.
- Experience of the effect of the drug, assessed using questionnaire PGIC score of 1–7.
- Control of vitamin $B_{12}$ is done by measuring cobalamin in plasma.
- Qualitative in-depth interviews will be conducted to capture women's lived experiences of pain, health, suffering and well-being.

### Cold pressor test

The cold pressor test[29–31] is measured in number of seconds the participant leaves her hand in the low-temperature water of 5°. Evaluation variables from the cold pressor test become pain threshold (when it starts to hurt) and tolerance time, that is, how long the participant endures the pain. The participants end the test by raising their hands when they can no longer stand it or when the set end time of 3 min has elapsed.

### Pressure algometry test

The pressure algometry test[32 33] generates a mechanical pressure against specific points on the body: trapezius (shoulder), epicondyle (elbow), gluteal (outside the gluteal muscles) and medial knee (inside of the knee), which are all accepted in the diagnosis of fibromyalgia. Pressure algometry tests are measured in kilopascal. The trial manager interrupts the mechanical pressure when the participant expresses pain.

### Numerical Rating Scale

The participants rate their pain intensity on NRS[34] in connection with the cold pressor test when the participant takes her hand from the water, after 1 min and finally after 3 min. The participants estimate their pain intensity on a scale (NRS) from 0 to 10, where 0 corresponds to no pain at all and 10 corresponds to the worst possible pain. Pain rating is measured in the same way after the pressure algometry test.

The influence of expectation, desire for pain relief and pain variability will be measured before each injection using the same NRS as used for pain ratings. Ratings of expected pain levels will be obtained by asking patients 'What level of pain do you expect when this treatment starts to have an effect?' The NRS scale is anchored at the left by the descriptors 'no pain sensation' and at the right by 'the most intense pain sensation imaginable'. Ratings of desired pain relief will be obtained by asking patients 'How strong is your desire for pain relief?' The question will be anchored by the descriptors '0=no desire for pain relief' and '10=the most intense desire for pain relief imaginable'. Ratings of pain variability will be obtained by asking the participants their average pain intensity that week. These scales have been validated and used in previous studies.[36–40]

### Short-form MPQ

In this study short-form MPQ of 15 questions[27] is used, which distinguishes between the sensory–discriminatory and the affective and emotional aspects of the pain experience. Participants are asked to estimate their pain intensity and to describe their pain in predetermined words. They fill in the MPQ on all three measurement occasions.

### RAND-36

RAND-36 is a quality-of-life instrument[28] that aims to measure health-related quality of life, that is, physical, mental and social well-being, not just the absence of illness. The estimation instrument consists of 36 questions of various kinds, for example, on general health, activity, physical health, emotional problems and pain. RAND-36 is filled in by the participant on all three measurement occasions.

### Patient Global Impression of Change

PGIC[35] evaluates participants' experience of the treatment, that is, the substance given. Participants tick the box that most closely describes the change that they are experiencing. The PGIC is completed by the participant after the injection treatment has been concluded at 12 weeks and at the follow-up 12 weeks later. PGIC is directly translated in its entirety from English by the research group.

### Cobalamin/p

Cobalamin/p levels are taken on three occasions. The first measurement, baseline, rules out a vitamin $B_{12}$ deficiency (according to the accepted reference values). A second measurement is performed at 12 weeks on completion of treatment in order to monitor the participants' level of cobalamin/p just after the end of the injection treatment. The final measurement is 12 weeks after the end of intervention.

### Qualitative in-depth interview

In order to obtain a deeper understanding of women's perceived experiences, in-depth interviews with both compliant and non-compliant participants from both groups will be performed in connection with the third measurement session. The aim is to describe the women's lived experiences of pain, health, suffering and well-being. The interviews will be conducted by the trial manager, recorded, transcribed and later analysed using an RLR (Reflective Lifeworld Research) approach.[24]

### Sample size

In order to obtain an adequate sample size, a power analysis was performed based on results from a previously published study[29] in which women with fibromyalgia

underwent the same stimulation procedure with the cold pressor test. Account is taken here of the SD shown and compensation for any placebo effects. Based on these results, the expected effect difference is estimated at 20 s with an SD of 20 s between the two groups after 12 weeks of treatment with mecobalamin (vitamin $B_{12}$)/placebo. Using a two-sided t-test for two independent groups with a power of 80% and 5% significance level, it was calculated that each group would consist of 16 participants. In order to compensate for any loss, 20 participants per group are therefore included. The study continues until 40 participants have been included and have completed the study.

### Randomisation and allocation concealment

Participants are randomised into two groups, the placebo group or the treatment group, each consisting of 20 participants either receiving mecobalamin (vitamin $B_{12}$) or placebo. An independent statistician generates a randomisation list by random distribution of processing in blocks (1:1) in a computerised statistics programme (STATA, version 17.1). Participants are randomised according to the list by the chief examiner opening closed randomisation envelopes in consecutive order. Based on each randomisation envelope, the chief examiner registers in and signs the injection retrieval protocol as to which substance the participant will receive. The participants' randomisation number is recorded in the journal. Only the chief examiner has access to randomisation envelopes and injection retrieval protocols, which are kept locked.

When the third measurement session has been completed, the trial manager contacts the chief examiner who has the randomisation list.

The chief examiner breaks the code and notifies the participant with a letter of given treatment. The code is broken at this time because it cannot be considered ethical that the participant does not find out about the treatment given, especially if the participant believes that the treatment has helped. If the participants have received mecobalamin and wish to continue, they are recommended to contact their health centre for continued mecobalamin treatment. The letter will contain the contact information of the trial manager and the chief examiner if the health centre doctor wishes to contact them. The trial manager is still unaware of what the individual participant received until the study is completed. If the code needs to be broken for an individual participant before the third measurement session has been completed, there is a code breaking envelope that may be used by the chief examiner in cases of emergency.

### Adverse events (AES) and serious adverse events (SAEs)

AEs are defined as any unfavourable or unintended reaction occurring in a research person during the course of the study and where investigational medicinal products have been administered, regardless of dose. Any AE will be reported from day 1 of the study at each visit to the clinic and until the last visit made 24 weeks after the start of the study. All SAEs that occur during the course of the study will be reported to the sponsor within 24 hours of the research staff becoming aware of the incident. During the study, an annual safety report from the sponsor and responsible investigator is sent in to both the Swedish Medical Products Agency and the regional ethical review board.

### Statistical methods

The primary endpoint is tolerance time at cold pressor test and will be calculated on the difference between the two groups after 12 weeks and analysed by analysis of variance (ANOVA). Pain thresholds measured by pressure algometry test (Somedic) will be analysed using t-test and ANOVA, similar to the primary variable. Questionnaires (MPQ, RAND-36 and PGIC) will be analysed using Mann-Whitney tests. In cases of minor loss (less than 10%), a mixed model can be used as an alternative to t-test and ANOVA with repeated measurement. For variables with more than 10% loss, imputation of data will be used. Drop-out participants will be described in a specific analysis. Rejection analysis will be performed on the participants who interrupt the study prematurely.

Results will be reported for both intention-to-treat and per-protocol analyses. Intention-to-treat is defined as the participant is placed in the treatment group he or she had been randomised to. Per-protocol is defined as the participant is placed in the treatment group she de facto received and carried out without significant protocol deviations. The per-protocol results will be presented as sensitivity analyses. The participant may miss a maximum of two injections.

Self-reported ratings of desired and expected treatment outcome pretreatment and post treatment will be implemented in linear regression models to predict any of the outcome measures. Repeated ratings of desired treatment outcome, expected treatment outcome and pain variability will be calculated as each participant's SD from the repeated ratings and implemented in a linear regression model to predict possible outcome measures.

### Patient and public involvement

There is no patient or public involvement in this study.

### Ethics and dissemination

The study is registered at the Swedish Medical Products Agency (EudraCT-no 2015-005086-23, appendix 5.1-2020-71076, protocol no 2020-08-17 V.5.0 and ClinicalTrials.gov). The trial was approved by the local ethical committee at Linkoping (EPM; 2018/294–31, appendices 2019–00347 and 2020–04482). The principles of the Helsinki Declaration[41] are followed regarding oral and written consent to participate, confidentiality and the possibility to withdraw participation from the study at any time. Good clinical practice is followed, with independent regular monitoring by Forum Ostergotland and Linkopings University, including plans for communicating important protocol modifications. The data will be placed in a locked cabinet at the university until the study is completed and the data are to be processed and

analysed. Only involved researchers have access to data when they are to be analysed; however, anyone requesting metadata can access the data during 2025–2035. The study started on 6 February 2019, but recruitment was delayed due to the COVID-19 pandemic. Final recruitment is expected to be completed in 2024. The results will primarily be communicated through peer-reviewed journals and conferences.

**Author affiliations**
[1]Department of Health and Caring Sciences, Linnaeus University, Linnaeus University Faculty of Health and Life Sciences, Växjö, Sweden
[2]Faculty of Health and Life Sciences, Centre of Interprofessional Cooperation within Emergency care (CICE), Linnaeus University, Växjö, Sweden
[3]Department of Research and development, Region Kronoberg, Vaxjo, Sweden
[4]The Pain Neuroimagine Lab, Karolinska Institute, Stockholm, Sweden
[5]Pain and Rehabilitation Centre, and Department of Medical and Health Sciences, Linkopings Universitet, Linkoping, Sweden

**Acknowledgements** We thank the coexaminer and medical doctors, Anders Willstedt and Bjarne Sorensen, as well as the registered nurses Ann-Louise Breider and Lisa Gunnarsson for their ongoing contributions and support.

**Contributors** KSH (trial manager), GL, KS, JF (chief examiner), MP, KJ, BG and CE (sponsor) planned the study design. KSH, GL, KS, AW and CE prepared the first version of the manuscript. All authors participated in the production of the analysis plan and in reviewing and drafting the manuscript, and read and approved the final version.

**Funding** This trial was supported by grants from Medical Research Council of Southeast Sweden (numbers 479961, 656241 and 854911) and Capio Research Foundation, Sweden (numbers 2018-3195 and 2019-3277). The funding sources had no involvement in study design and data collection or the writing of this manuscript.

**Competing interests** None declared.

**Patient and public involvement** Patients and/or the public were not involved in the design, conduct, reporting or dissemination plans of this research.

**Patient consent for publication** Consent obtained directly from patient(s).

**Provenance and peer review** Not commissioned; externally peer reviewed.

**ORCID iD**
Carina Elmqvist http://orcid.org/0000-0001-8376-8805

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
