## [Reviewer comments · BMJ Open]

ARTICLE DETAILS

TITLE (PROVISIONAL)	Efficacy of mecobalamin (vitamin B12) in the treatment of long-term pain in women diagnosed with fibromyalgia: protocol for a randomized, placebo-controlled trial
AUTHORS	Sall Hansson, Karin; Lindqvist, Gunilla; Stening, Kent; Fohlman, Jan; Wojanowski, Anna; Ponten, Moa; Jensen, Karin; Gerdle, Björn; Elmqvist, Carina

VERSION 1 – REVIEW

REVIEWER	Adıgüzel, Kübra Tel University of Health Sciences Turkey
REVIEW RETURNED	06-Sep-2022

GENERAL COMMENTS	This is an interesting study and i think that the results will be valuable. Here, i have some minor questions. 1. How will you control or document dietary B12 intake? I recommend to obtain 24-hour dietary record weekly and compare dietary intake. 2. Is there a physical activity record in the study protocol? What if some of the included patients perform exercise or quit their routine exercise program? 3. In abstract section: do not use abbreviation for 'QST measured'. Because it is not repeated in abstract section.
--

REVIEWER	Goni, Vijay Post Graduate Institute of Medical Education and Research
REVIEW RETURNED	05-Oct-2022

GENERAL COMMENTS	It is a well constructed study protocol. This should be accepted and pursued as intended.
---

REVIEWER	Ghavidel-Parsa, Banafsheh Guilan University of Medical Sciences
REVIEW RETURNED	04-Dec-2022

GENERAL COMMENTS	It is very interesting study protocol about the effect of B12 on pain in fibromyalgia patients. There are some important issues that help to achieve more real results: 1. Considering heterogeneous spectrum of fibromyalgia patients and variable response to treatment in fibromyalgia, it would be reasonable to incorporate some severity scales such as FIQR or PSD for subanalysis of data and categorization of patients into mild, moderate and severe impact. 2. The authors mentioned this study as the first trial of B12 on fibromyalgia. I refer you to the recent publication (2022) by Gharibpoor et al that investigated the effect of B12 on fibromyalgia
---

	patients as an pre-post study. Although this study was not the randomized-controlled trial, it showed the promising results about use of B12 in fibromyalgia patients. 3.I did not see any study limitaions in the protocol. The low numbers of participants, heterogenesity of fibromyalgia spectrum, large placebo effects of parantral injection (despite controlled group), and so on can be the important limitaions.
--	--

VERSION 1 – AUTHOR RESPONSE

Reviewer 1 comments	
This is an interesting study and i think that the results will be valuable. Here, I have some minor questions.	Thank you
How will you control or document dietary B12 intake? I recommend to obtain 24-hour dietary record weekly and compare dietary intake.	Thank you, it's a brilliant idea for future studies, however in this study we do not intend to control or document the dietary B12 intake. Oral intake would not lead to the high concentration needed to penetrate into CNS. However, we measure cobalamin/s three times during the study.
Is there a physical activity record in the study protocol? What if some of the included patients perform exercise or quit their routine exercise program?	We have no special physical activity record but questions about physical activity is recorded in RAND-36 three times during the study.
In abstract section: do not use abbreviation for 'QST measured'. Because it is not repeated in abstract section.	Thank you, the abbreviation is now deleted in the abstract
Reviewer 2 comments	
It is a well constructed study protocol. This should be accepted and pursued as intended.	Thank you
Reviewer 3 comments	
It is very interesting study protocol about the effect of B12 on pain in fibromyalgia patients. There are some important issues that help to achieve more real results:	Thank you
Considering heterogeneous spectrum of fibromyalgia patients and variable response to treatment in fibromyalgia, it would be reasonable to incorporate some severity scales such as FIQR or PSD for subanalysis of data and categorization of patients into mild, moderate and severe impact.	Thank you, In the start of the study, we decided to use RAND-36 Health and life quality, McGills questionnaire short form and Patients' Global Impression of change (PGIC). All these assessments measure the impact on the patients.
The authors mentioned this study as the first trial of B12 on fibromyalgia. I refer you to the recent publication (2022) by Gharibpoor et al that investigated the effect of B12 on fibromyalgia patients as a pre-post study. Although this study was not the randomized-controlled trial, it showed the promising results about use of B12	Thank you for an interesting publication which now is included in the introduction of the manuscript.

in fibromyalgia patients.	
I did not see any study limitations in the protocol. The low numbers of participants, heterogeneity of fibromyalgia spectrum, large placebo effects of parenteral injection (despite controlled group), and so on can be the important limitations.	Thank you, limitations are now included and clarified in the main document

VERSION 2 – REVIEW

REVIEWER	Adıgüzel, Kübra Tel University of Health Sciences Turkey
REVIEW RETURNED	05-Mar-2023

GENERAL COMMENTS	I reviewed the revisions. Best regards.
---

REVIEWER	Ghavidel-Parsa, Banafsheh Guilan University of Medical Sciences
REVIEW RETURNED	06-Feb-2023

GENERAL COMMENTS	Thank you for your corrections
--------------------------------